# Long Non-Coding RNA Regulation of Epigenetics in Vascular Cells

**DOI:** 10.3390/ncrna7040062

**Published:** 2021-09-23

**Authors:** Hester Garratt, Robert Ashburn, Miron Sopić, Antonella Nogara, Andrea Caporali, Tijana Mitić

**Affiliations:** 1Centre for Cardiovascular Science, The Queen’s Medical Research Institute, Edinburgh BioQuarter, 47 Little France Crescent, Edinburgh EH16 4TJ, UK; s1612121@sms.ed.ac.uk (H.G.); robert.ashburn@imba.oeaw.ac.at (R.A.); anogara@exseed.ed.ac.uk (A.N.); 2Department of Medical Biochemistry, Faculty of Pharmacy, University of Belgrade, 11000 Belgrade, Serbia; miron.sopic@pharmacy.bg.ac.rs

**Keywords:** epigenetics, lncRNA, PRC2, vascular biology, chromatin, RNA-protein

## Abstract

The vascular endothelium comprises the interface between the circulation and the vessel wall and, as such, is under the dynamic regulation of vascular signalling, nutrients, and hypoxia. Understanding the molecular drivers behind endothelial cell (EC) and vascular smooth muscle cell (VSMC) function and dysfunction remains a pivotal task for further clinical progress in tackling vascular disease. A newly emerging era in vascular biology with landmark deep sequencing approaches has provided us with the means to profile diverse layers of transcriptional regulation at a single cell, chromatin, and epigenetic level. This review describes the roles of major vascular long non-coding RNA (lncRNAs) in the epigenetic regulation of EC and VSMC function and discusses the recent progress in their discovery, detection, and functional characterisation. We summarise new findings regarding lncRNA-mediated epigenetic mechanisms—often regulated by hypoxia—within the vascular endothelium and smooth muscle to control vascular homeostasis in health and disease. Furthermore, we outline novel molecular techniques being used in the field to delineate the lncRNA subcellular localisation and interaction with proteins to unravel their biological roles in the epigenetic regulation of vascular genes.

## 1. The Role of the Vascular Endothelium in Health and Disease

The vascular endothelium comprises the layer of endothelial cells (ECs) that create the interface between the circulation and the vessel wall that is under dynamic regulation in accordance with its environment to maintain perfusion and the supply of oxygen and nutrients required for cellular metabolism [1,2]. A healthy vascular endothelium is key to maintaining vascular homeostasis, but little is known of what regulates its distinct and characteristic function. The angiogenic capacity of ECs is vital during embryonic and vascular tissue development and during wound healing in damaged tissues. Homeostatic mechanisms encourage hypoxia- or vascular endothelial growth factor (VEGF)-induced angiogenesis required for inflammatory and reproductive processes and post-ischaemic tissue regeneration (Figure 1A); and suppress vascular inflammation and thrombosis [3,4,5]. The vasculature is implicated in a range of pathological disease states, such as coronary and small vessel disease, peripheral arterial diseases, hypertension, aortic aneurysm, and diabetes mellitus—all of which are worldwide leading causes of morbidity and mortality [6]. Endothelial cell dysfunction can occur due to hypoxic or oxidative injury and predisposes to atherosclerotic and ischaemic disease by potentiating vascular inflammation and atherothrombosis (Figure 1B). Hence, an increasing understanding of endothelial homeostasis is continually sought as a primacy to the comprehension of many complex cardiovascular conditions. Increasingly, studies are being designed to elucidate molecular mechanisms behind EC and VSMC dysfunction and investigate drug therapies which induce revascularisation to facilitate repair of ischaemic sites. 

## 2. The Era of lncRNAs in Vascular Biology

A new era for vascular biology has emerged with technological advancements and next-generation sequencing (NGS) approaches which have largely empowered our exploration of genomes over the past ten years [8]. Hence, the focus has shifted from the coding to the non-coding part of the genome, comprising non-coding RNA genes, regulatory DNA sequences, introns, and interspersed nuclear elements. The large number of non-coding RNAs named short (microRNA, miRNA) and long non-coding RNAs (lncRNAs) regulate important cellular functions—including within vascular cells. Beside the ENCODE database (https://github.com/ENCODE-DCC/encoded/releases/tag/v121.0, accessed on 4 May 2021), there is now NONCODE (http://www.noncode.org/, accessed on 8 August 2021), an integrated knowledge database dedicated especially to lncRNAs [9]. 

The lncRNAs are a heterogeneous group of RNAs defined as transcripts >200 nucleotides long. LncRNAs can be categorised by their origin of transcription and position of neighbouring genes into sense, natural antisense, intronic, intergenic, bidirectional promoter, and enhancer lncRNAs. Their biogenesis is distinct from that of mRNA as they are poorly evolutionarily conserved, often less abundant, more tissue-specific, display distinct spatiotemporal expression and specific subcellular localisation and function, and tend to form secondary structural domains folded into specific scaffolds [10,11]. Indeed, despite significant differences in the primary sequence, lncRNA homology across species can be identified through the analysis of their secondary and tertiary structures key to their specific function [12]. The use of NGS combined with high-resolution methods to detect lncRNA sites of interaction on a genome or with other RNA species (via enzymatic or chemical probing) has provided insight into the structural folding modules, domains, and loops for several vascular lncRNAs like Braveheart, HOTAIR, MEG3, RepA, Xist, and SRA [13]. Their structural conservation, as opposed to nucleotide sequence conservation, appears to be crucial for maintaining their function. For example, maternally expressed gene 3 (MEG3) is a highly structured endothelial-enriched lncRNA whose complex secondary and tertiary molecular architecture permits its key interactions with epigenetic regulators and tumour suppressor genes [14,15,16]. The tertiary structure of MEG3—possibly owing to conserved motifs with a high GC (guanine-cytosine) content (57%)—enables the formation of pseudoknots that are crucial to its regulation of the p53 pathway, DNA damage response, and associated apoptosis to protect EC function [14,17,18]. Understanding of lncRNA functions in vivo is additionally complicated by the fact that, due to alternative splicing, lncRNAs can exist in multiple functional isoforms in the cell. For example, different transcripts of antisense ncRNA in the INK4 locus (ANRIL) have opposite expression patterns in coronary arteries and even exert antagonising functions in ECs [17].

Recently, greater focus has been placed on identifying the subcellular localisation of vascular lncRNAs in order to functionally characterise nuclear-, cytoplasmic-, and chromatin-enriched ones [18]. Indeed, MEG3, MALAT1 (metastasis associated lung adenocarcinoma transcript 1), NEAT1 (nuclear enriched abundant transcript 1 or nuclear paraspeckle assembly transcript 1), LEENE (lncRNA that enhances eNOS expression), and GATA-6AS (antisense transcript of GATA6) in endothelial cells, and ANRIL, SMILR (smooth muscle enriched long noncoding RNA), and SENCR (smooth muscle and endothelial cell enriched migration/differentiation-associated lncRNA) in vascular smooth muscle cells have been reported as regulators of endothelial and smooth muscle differentiation [1,8]; vascular cell proliferation, apoptosis, and vascular remodelling [19,20]; and endothelial function and angiogenesis [5,21,22,23,24,25] through affecting gene transcription to mediate endothelial homeostasis or lead to vascular pathologies (Table 1).

ENCODE RNA sequencing datasets suggest that subcellular lncRNA nuclear/cytoplasmic ratios may be mechanised by factors including slow nuclear export, nuclear or cytoplasmic degradation, and splicing efficiency [32]. Increasingly, lncRNAs are reported to be retained for activity in the nucleus (nucleolus and chromatin) and associated with local functions including mediating inter- and intra-chromosomal interactions and sequestering RNA complexes at chromatin [33]. Through their primary sequence or secondary structures, nuclear lncRNAs can transcriptionally or post-transcriptionally moderate vascular gene expression through cis- or trans-regulatory mechanisms. LncRNAs may regulate interactions with DNA/RNA-binding proteins at specific loci through acting as guides, scaffolds, or decoys, may mediate chromatin looping and higher-order chromatin interactions between promoters and distal enhancers, and may control alternative splicing, whilst lncRNA transcription itself may favourably re-shape chromatin confirmation, aid transcription factor recruitment, or impose enhancer-like effects on the neighbouring genes through the transcription from enhancer regions (aka elncRNAs) [34,35].

Cytoplasmic localisation, however, allows for lncRNA modulation of mRNA stability and translation, microRNA activity, protein turnover, and signal transduction pathways through acting as cytosolic scaffolds, degradation tags, sponges, and molecular decoys—ultimately mediating physiological and pathological vascular processes, including angiogenesis and atherosclerosis in diseases such as hypertension, retinopathy, and malignancy [36,37]. MEG3, for example, has been observed to translocate to the cytoplasm upon hypoxia in pulmonary artery smooth muscle cells (PASMCs), subsequently sequestering and inducing the degradation of microRNA miR-328. This ultimately upregulates IGF1R and regulates hypoxia-induced PASMC proliferation and pulmonary hypertension [38]. 

In addition to intracellular localization, lncRNAs have been detected extracellularly in several body fluids, including whole blood, plasma, urine, saliva, and gastric juice [39,40,41]. Extracellular lncRNAs are relatively resistant to degradation, reflect dynamically intracellular changes during disease, and are easily detected in body fluids; thus, showing high biomarker potential [38]. Diagnostic and prognostic properties of lncRNAs have been demonstrated in cardiometabolic diseases. The plasma levels of mitochondria-derived lncRNA long intergenic non-coding RNA predicting cardiac remodelling (LIPCAR) are associated with left ventricular remodelling following myocardial infarction and an increased risk of developing heart disease [42]. ANRIL and QSOX1 (quiescin sulfhydryl oxidase 1) have also been detected in plasma of such individuals and thereby serve as potential biomarkers—although it is unclear which cellular compartment or cell type they arise from. Moreover, circulating SENCR levels have been shown to discriminate pioglitazone therapy responders from non-responders to a better extent than traditional clinical markers [43,44]. Many other circulating lncRNAs may prove to be powerful biomarkers for diabetic and cardiovascular disease states, yet their diagnostic and prognostic values remain an open field of investigation [40].

Remarkably, some recent studies have even questioned the non-coding nature of lncRNAs. Novel in silico approaches have suggested the existence of small open reading frames (ORFs) within a small number of lncRNAs, whilst ribosomal profiling has revealed their potential to be translated [45]. Mass spectrometry-based proteomics approaches coupled with transcriptome sequencing have effectively been used to identify micro-peptides encoded by putative lncRNAs. The validation of ORFs and post-translational function is required to establish any roles these micro-peptides may have in vascular biology [46].

One resource that has helped vascular researchers explore the layers of gene regulation to assess direct and controlled comparison of gene expression between different cell types is the Tabula Muris, available here https://tabula-muris.ds.czbiohub.org/, accessed on 6 June 2020) [47]. Similarly, the single-cell transcriptional profiles of fibroblasts and the vascular mural cell are valuable resource serving as a “postal code” to provide molecular information to vascular researchers, available here http://betsholtzlab.org/Publications/databases.html, accessed on 2 September 2020) [48]. RNA sequencing data of the brain, lung, and heart endothelial translatome from adult Cdh5CreERT2^/+^; Rpl22HA^/+^ mice at baseline and during inflammation can be visualised at http://rehmanlab.org/ribo [2] (accessed on 4 May 2021); the human heart translatome visualised at https://shiny.mdc-berlin.de/cardiac-translatome/; [49], (accessed on 4 May 2021) and the hypoxia-induced rat cardiomyocytes translatome visualized in reference [50].

## 3. Long Non-Coding RNAs as Remodellers of the Vascular Epigenome

Epigenetics imply heritable changes to the genome that do not involve changes to the underlying nucleotide sequence. Although the biological relevance for the vast majority of lncRNAs remains elusive, lncRNAs are known to influence gene expression at a transcriptional, post-transcriptional, translational, and post-translational level; and to represent a new layer of epigenetic control in vascular homeostasis that is dysregulated in disease [6,20,47]. 

Epigenetic changes fine-tune vascular gene expression in a temporal, tissue-specific fashion through functional chromatin alterations which alter DNA accessibility at specific loci for replicative enzymes or transcription factors. The mechanisms include (a) DNA methylation, (b) histone tail modification, and (c) chromatin remodelling. DNA methylation typically acts to repress gene expression through impeding transcription factor binding; histone-modifying enzymes (“writers” and “erasers”) catalyse the addition and removal of acetylation and methylation marks to alter chromatin architecture; whilst chromatin remodellers re-position nucleosomes to condense or decondense chromatin, restricting or enabling access for transcriptional machinery, respectively (Figure 2A–C).

Paragons of endothelial-enriched genes crucial to endothelial homeostasis, such as *eNOS* (endothelial Nitric Oxide Synthase) and many pro-angiogenic genes implicated in the revascularisation response to hypoxia, are known to be dynamically epigenetically regulated by DNA methylation and changes in the histone density [51]. Moreover, histone methylation has been shown to epigenetically regulate gene expression in accordance with a hypoxic environment. Jumonji C domain-containing histone demethylases are part of the 2-oxoglutarate-dependent dioxygenase (2-OGDD) enzyme family which depend on oxygen as a co-factor for their activity. In the absence of oxygen, dioxygenase activity, and therefore lysine demethylase activity, is inhibited—inducing histone tail hypermethylation and altering the expression of certain genes to control hypoxia-dependent cell fate [52,53]. 

LncRNAs can act via multiple mechanisms to epigenetically regulate vascular gene transcription—including acting to recruit, scaffold, or sequester such chromatin remodellers and chromatin modifiers to influence chromatin reorganisation (Table 1) [6]. Some lncRNAs can be taken up by extracellular vesicles (EVs) as exosomes that are secreted by many cell types as cell–cell communicators in response to stress conditions. Hence, lncRNA-enriched EVs are readily being reported as biomarkers [54]. The active process of lncRNA sorting into exosomes can be facilitated by RNA-binding proteins (RBPs) with a growing understanding of their mechanism of loading and action involving epigenetic modifications, through which exosomal lncRNAs can alleviate the progression of some diseases [55,56].

Further examples of lncRNA-mediated epigenetic regulation in vascular cells, including those interacting with one of the major transcriptional repressors, polycomb repressive complex 2 (PRC2), are discussed below.

## 4. Polycomb Repressive Complex 2 and lncRNAs

The polycomb repressive complex 2 (PRC2) is a major histone methyltransferase in vascular biology. Through the deposition of histone H3 lysine K27 trimethylation (H3K27me3) modification, PRC2 compacts chromatin and represses gene expression. PRC2 comprises four major subunits including EZH2 (enhancer of zeste homolog 2) which possesses the catalytic histone methyltransferase SET domain; EED (embryonic ectoderm development) which binds the H3K27me3 mark and is responsible for its propagation; SUZ12 (suppressor of zeste 12 homolog); and RbAp46/48 (retinoblastoma protein associated protein 46/48) (Figure 3) [57]. The mechanisms behind PRC2 engagement with native chromatin, as well as its recruitment in a context-dependent manner, remain to be fully understood [58]. EZH2-mediated epigenomic landscapes guide cardiovascular and endothelial differentiation and maturation, maintain tissue-specific genetic blueprints established during development, and are implicated in cardiovascular disease [55,57,58]. Moreover, PRC2 histone methyltransferase activity appears to be a key regulator of angiogenesis. An increased EZH2 expression negatively regulates the revascularisation response to hypoxia through the deposition of H3K27me3 onto pro-angiogenic genes such as *eNOS,* thus worsening outcomes in ischaemia, and similarly halts the endothelial lineage commitment in pro-angiogenic haematopoietic and endothelial progenitor cells [20,48,59].

It is now recognised that in addition to its chromatin binding capacity, PRC2 displays a tissue-specific lncRNA-binding capacity, including an ability to bind lncRNAs involved in vascular endothelial homeostasis such as MEG3—elucidating a model for PRC2 recruitment to specific loci. Both EZH2 and SUZ12 subunits bind lncRNA through conserved sequences and interact with chromatin in a mutually antagonistic fashion, with subsequent lncRNA-mediated target gene recognition through triplex formation—representing a general mechanism to guide transcriptional regulators [59]. As well as serving as a scaffold to mediate H3K27me3 deposition on chromatin, motif-driven RNA binding may fine-tune transcriptional repression by recruiting PRC2 to stall the activity of RNA polymerase-II [60]. 

## 5. Multi-Omics Approaches to Detect Long Non-Coding RNAs in Vascular Cells

The use of novel technologies has immensely improved the detection of lncRNAs and the identification of their in vivo targets [61]. Chromatin immunoprecipitation (ChIP) and assay for transposon-accessible chromatin (ATAC-seq) allow the mapping of protein-DNA interaction in vivo on a genome scale [62]. RNA immunoprecipitation (RIP), chromatin isolation by RNA purification (ChIRP), individual-nucleotide-resolution UV cross-linking and immunoprecipitation (iCLIP), and cross-linking, ligation and sequencing of hybrids (CLASH) use glutaraldehyde-, formaldehyde- and UV-mediated cross-linking approaches in order to preserve RNA–RNA and RNA–protein interactions and to capture specific lncRNAs and their distribution between intracellular fractions using biotinylated antisense DNA probes [63,64,65].

In Table 1, we have outlined the experimental approaches used to delineate epigenetic functions of some nuclear lncRNAs in a vascular setting, whilst Table 2 reports the experimental use of these techniques in the vascular field between 2018 and present. The techniques for mapping the RNA interactome have also been comprehensively reviewed recently (see reference [33]).

### 5.1. MEG3

Maternally expressed gene 3 (MEG3) is a major hypoxia-sensitive lncRNA which interacts with PRC2 and has complex roles in EC function. Hypoxia-inducible factor 1α (HIF1α) upregulates the MEG3 expression in response to hypoxia [66] and under hypoxic conditions, MEG3 has been observed to translocate to the cytoplasm [38]. MEG3 contains a 356-nucleotide element which associates with U1 small nuclear ribonucleoprotein particles (snRNP) and is responsible for its nuclear retention [67]. MEG3 is a negative regulator of angiogenesis in hypoxia, with *MEG3* knock-out mice showing enhanced angiogenesis in the brain and the augmented expression of pro-angiogenic genes [68]. MEG3 is also, however, required for VEGF-induced EC migration and angiogenesis in human umbilical vein endothelial cells (HUVECs) [64]. It has been proposed that MEG3 both negatively regulates pro-angiogenic genes and contributes to age-associated endothelial dysfunction through its interaction with EZH2 or JARID2 and the recruitment of PRC2 to epigenetically regulate chromatin—yet these mechanisms are yet to be fully elucidated [24]. Indeed, it is known that MEG3 modulates the activity of genes in the transforming growth factor-b (TGF-b) pathway through its interaction with PRC2 and formation of RNA–DNA triplexes [26].

### 5.2. MALAT1

LncRNA-metastasis-associated lung adenocarcinoma transcript 1 (MALAT1) has been implicated in the development of diabetic nephropathy through its epigenetic regulation of renal glomerular EC (HRGEC) gene expression [27]. It was demonstrated that MALAT1 was upregulated in patients with diabetic nephropathy and inversely correlated with the expression of klotho—a gene known to improve the function of hyperglycaemia (HG)-induced glomerular EC injury. Authors used RIP and ChIP approaches in HRGECs following HG-induced injury to define the underlying mechanism. MALAT1 mediates HG-induced HRGEC injury through the binding and recruitment of G9a methyltransferase to deposit the H3K9me1 repressive signature to epigenetically suppress *klotho* expression. Such research not only demonstrates the role of MALAT1 in devising the chromatin status of *klotho* but implies the prospect of targeting MALAT1 to ameliorate glomerular EC function in hyperglycaemia to impede the progression of diabetic nephropathy. 

Epigenetic pathways regulated by MALAT1 are also implicated in the development of thoracic aortic aneurysm (TAA) and stenotic disease. The dysfunction of the vascular smooth muscle cell (VSMC) through deficiencies in contractile proteins predisposes to the pathogenesis of these conditions. Histone deacetylase complex, HDAC9, is known to recruit EZH2 to epigenetically modify chromatin through the deposition of repressive H3K27me3. EZH2 was identified as a crucial player in TAA development, with its inhibition-enhancing expression of contractile protein genes such as *SM22a* and improving aortic performance in mice [29]. Moreover, pathological recruitment of a nuclear HDAC9–MALAT1–BRG1 (brahma-related gene-1) complex to deposit H3K27me3 on gene promoters and downregulate contractile protein gene expression has been highlighted as a pathway implicated in VSMC dysfunction in TAA. Indeed, *Malat1* or *Hdac9* disruption restored a contractile phenotype, ameliorated aortic wall architecture, and prevented experimental TAA growth [28]. MALAT1-mediated nuclear targeting of HDAC9 to transcriptionally silence genes encoding contractile proteins also mediates stenotic vascular disease, with MALAT1 or HDAC9 deficiency or EZH2 inhibition attenuating neointimal formation and preserving the beneficial VSMC contractile phenotypes [29]. 

### 5.3. NEAT1

Although nuclear enriched abundant transcript 1 (NEAT1) is nuclear lncRNA, the findings have shown that its localisation is dispersed within the nucleus and is necessary for the formation of paraspeckles [69]. Ahmed et al. demonstrated that lncRNA-NEAT1 expression was upregulated in VSMCs in response to vascular injury to promote a proliferative phenotype and blood vessel repair [22]. NEAT1 silencing in human coronary artery smooth muscle cells (HCASMCs) increased SM-specific gene expression and reduced VSMC proliferation and migration; whilst *NEAT1* knockout (KO) mice displayed decreased neointima formation and vascular repair. Moreover, ChIP assays showed that *NEAT1* KO in HCASMCs enriched active chromatin marks (H3K4me3 and H3K9ac) and depleted inactive modifications [22]. Mechanistically, RIP showed binding of NEAT1 to the WDR5 chromatin modifier which has a scaffolding role for H3K4 methyltransferase complexes to deposit the H3K4 mark and contribute to an open chromatin state. Upon stimulation of a proliferative, anti-contractile phenotype, the NEAT1 binding to WDR5 was enhanced. Overall, it appears that NEAT1 operates by a mechanism of sequestering WDR5 from SM contractile genes to initiate epigenetic repression of SM contractile gene expression and modulate phenotypic switching towards a proliferative phenotype. 

### 5.4. MANTIS

MANTIS is an intronic lncRNA-located antisense to Annexin A4 (*ANXA4*) that acts as an important regulator of angiogenic sprouting and alignment of ECs [23]. The downregulation of MANTIS expression, controlled by histone demethylase JARID1B, was observed in the lungs of patients with idiopathic pulmonary arterial hypertension, and the upregulation was observed in carotid arteries of *Macaca fascicularis* under an atherosclerosis regression diet, and in endothelial cells isolated from human glioblastoma patients. Knocking out MANTIS with LNA-GapmeRs led to the downregulation of key endothelial genes important for angiogenesis including *SOX18*, *SMAD6*, and *COUP-TFII,* and the upregulation of stress-induced genes contributory to atherosclerosis such as interleukin 6 *(IL-6)* and superoxide dismutase 2 *(SOD2).* Genes in proximity to the MANTIS locus in the genome were not influenced by the MANTIS knockout, suggesting that it acts in a trans- rather than cis-regulatory manner. Mechanistically, MANTIS appears to influence epigenetic gene regulation through acting via the BRG1 component of the chromatin remodelling complex, SWI/SNF (SWItch/Sucrose non-fermentable), to facilitate access by RNA polymerase II machinery to the specific genes which counteract atherosclerotic processes [23].

MANTIS appears to further impede atherosclerosis development by limiting pro-inflammatory intercellular adhesion molecule 1 (ICAM-1)-mediated endothelial monocyte adhesion by diminishing binding of BRG1 at the *ICAM-1* promoter to repress its transcription. Indeed, MANTIS silencing has been shown to increase endothelial monocyte adhesion via ICAM-1. Moreover, statins and laminar flow ameliorate atherosclerotic vascular disease by inducing MANTIS expression through epigenetic mechanisms and Krüppel-like factor (KLF) 2 and 4 transcription factors [70]. Since MANTIS appears to both limit atherosclerosis development in a multitude of ways and mediate the pleiotropic effects of statins, it may be an effective therapeutic target for ameliorating vascular disease.

### 5.5. LEENE

LncRNA that enhances eNOS expression (LEENE) is an enhancer-associated RNA that plays an important role in regulating *eNOS* expression and endothelial barrier function [21,34]. Like many athero-protective genes, LEENE expression is regulated by two transcription factors, KLF2 and KLF4. Miao et al. used the chromatin conformation capture methods to demonstrate that LEENE targets the distal enhancer region located near the *eNOS* promoter in ECs, acting as a facilitator of RNA polymerase II recruitment to the *eNOS* promoter to enhance its transcription. LEENE RNA alone was not sufficient to enhance the eNOS transcription, yet genomic or transcriptional inhibition of the LEENE enhancer region suppressed the *eNOS* transcription and increased the transcription of pro-inflammatory and pro-atherosclerotic *ICAM1* and *VCAM1* [21]. Moreover, a cytoplasmic function has been described for LEENE (linc00520) comprising the regulation of pathways central for VEGF signalling and cell adhesion during the endothelial response to shear stress, resulting in its additional name, shear stress-induced long non-coding RNA (LASSIE) [71]. LASSIE associates with junctional proteins, such as PECAM-1, and the intermediate filament protein nestin, serving as a link between the cytoskeleton and adherent junctions to regulate EC alignment and vascular barrier function [71].

### 5.6. GATA6-AS

It has previously been shown that HIF1α-dependent signalling pathways promote the hypoxia-induced, pathology-associated cellular process of endothelial–mesenchymal transition (endMT) through mediating epigenetic changes via methyltransferases in endothelial cells [10]. Since a plethora of lncRNAs are known to respond to HIF [66] and together regulate methylase enzymes [72,73], it is unsurprising that further studies have illuminated a role for lncRNAs in the regulation of endMT. A study by Neumann et al. recently characterised the participation of the nuclear-enriched, hypoxia-regulated lncRNA antisense transcript of GATA6 (GATA6-AS) in the epigenetic regulation of endothelial gene expression underlying endMT [30]. Using an RNA deep sequencing approach, the authors identified that lncRNA GATA6-AS was upregulated under hypoxia and regulated endMT as well as sprouting angiogenesis and endothelial cell migration. GATA6-AS silencing impaired the endMT in vitro and augmented angiogenesis in vivo in a HUVEC-based xenograft model. Mechanistically, using the RIP and iCLIP techniques in HUVECs, the authors demonstrated that GATA6-AS associated with the LOXL2 deaminase enzyme, known to remove activating H3K4me3 marks, regulates the angiogenesis-related genes *periostin* and *cyclooxygenase-1*. Since GATA6-AS silencing reduced H3K4me3 of genes, this suggested that lncRNA GATA6-AS negatively regulates LOXL2 the demethylase function—promoting H3K4me3 mark deposition and the activation of angiogenesis-related genes.

### 5.7. ANRIL 

Studies are increasingly revealing the implication of lncRNA-mediated histone methylation in the progression of atherosclerosis. LncRNA-ANRIL (antisense ncRNA in the INK4 locus), otherwise known as cyclin-dependent kinase inhibitor 2B antisense transcript 1 (CDKN2B-AS1), is known to contribute to atherosclerotic disease states including stroke, coronary heart disease, and peripheral vascular disease through epigenetically regulating the endothelial inflammatory response as well as cell proliferation, adhesion, and apoptosis [31]. ANRIL promotes plaque formation due to its epigenetic regulation of chromatin regions in cis by recruitment of PRC1 and PRC2, and its ability to bind specific sequences on regulators such as CCCTC-binding factor (CTCF) to regulate distant genes in trans [31]. For example, ANRIL has been found to contribute to the formation of atherosclerotic plaques through epigenetically regulating the CDKN2B promoter [74]. ANRIL expression was found to be upregulated in atherosclerotic compared to non-atherosclerotic tissues and in macrophage-derived foam cells. In contrast, CDKN2B was down regulated. To elucidate the molecular mechanism of ANRIL, authors used a variety of techniques. RNA–DNA pull down and capture was used to identify RNA–DNA triplexes between nuclear-localised ANRIL and the CDKN2B promoter. RIP and ChIP approaches were used to determine interactions between ANRIL, CCCTC-binding factor (CTCF), EZH2, and CDKN2B. It was found that ANRIL regulated CDKN2B transcriptional repression through EZH2 recruitment and H3K27me3 deposition onto the CDKN2B promoter, under the regulation of CTCF. ANRIL silencing attenuated atherosclerosis in macrophage-derived foam cells and in ApoE^−/−^ mouse models of atherosclerosis and promoted the reversal by macrophage reverse cholesterol transport (mRCT)—suggesting a potential therapeutic modality in targeting ANRIL for treating atherosclerosis. 

## 6. Application and RNA Therapeutics

Our evolving understanding of lncRNAs and their role in epigenetic regulation of vascular cells may have a profound translational impact. Epigenetic therapies have already shown therapeutic potential in the context of cancer and PRC2 inhibitor agents such as EED226, EPZ, CPI-1205, and GSK126, which have also shown effectiveness in other settings. EZH2 inhibition led to the increased expression of pro-angiogenic genes and enhanced angiogenic functions and EC proliferation in vitro, whilst promoting blood flow recovery in mouse ischaemic muscles in vivo [51]. In a mouse model of TAA, EZH2 inhibition improved the outcomes due to the augmented expression of pathologically inactivated genes encoding contractile proteins [29]. The inhibition of EZH2 can also facilitate the activation of cardiac genes, indicating a promising way to generate or reprogramme human induced cardiomyocyte-like cells (hiCMs) [75].

Previously, when designing such PRC2 inhibitors, the RNA binding capacity of PRC2 was not taken into consideration. Disruption of lncRNA-mediated PRC2 recruitment has, however, been shown to reduce the severity of spinal muscular atrophy due to reduced H3K27me3 occupancy at the *SMN* gene, increasing its expression [76]. Moreover, another study indicated the use of a selective compound to manipulate the interaction between lncRNA-HOTAIR and EZH2 to mediate the downstream epigenetic mechanisms in the context of cancer therapy [77]. New therapeutic opportunities are therefore arising to block PRC2 recruitment effectively by dissociating lncRNAs using targeted therapy. With this novel class of pharmacological PRC2 inhibitors there is a potential to study the lncRNA epigenetic modulation of vascular gene expression and assess the therapeutic effectiveness of selective interference with the lncRNA-PRC2/EZH2 interaction [78].

Publications are increasingly noting the potential for lncRNA therapeutics in cardiovascular disease, with multiple vascular lncRNAs being identified as promising targets for the inhibition by antisense oligonucleotides and short hairpin RNAs in the treatment of atherosclerotic disease and in the induction of angiogenesis and vascular remodelling and repair [79,80]. In vivo GapmeR-mediated inhibition of MEG3, for example, reduced cardiac fibrosis, indicating its potential as a novel target for the prevention of cardiac remodelling, whilst a similar inhibition of MALAT1 attenuated the endothelial revascularisation response following ischaemia [25,80]. HOTAIR, guided by the transcriptional factor SNAIL, acts as a site-specific scaffold for PRC2 to mediate the repression of epithelial genes. Garbo et al. effectively counteracted HOTAIR function in the context of hepatocellular carcinoma cells with few off-target effects to impair the epithelial to mesenchymal transition (EMT) responsible for driving vascular inflammation [81]. Such research, in combination with our growing understanding of lncRNA-mediated epigenetic mechanisms in vascular cells, implies great potential for lncRNAs as targets to mitigate endothelial or vascular smooth muscle cell dysfunction, initiate therapeutic angiogenesis and vascular regeneration, and ultimately treat ischaemic cardiovascular diseases. 

RNA molecules are an attractive therapeutic concept due to their low immunogenicity, effective cell and nuclear penetration, low cost to manufacture, and ease of chemical modification [81]. Multiple challenges in the development, however, have been identified, which may limit the clinical application of RNA therapeutics. These include the difficulty in delivering high enough doses of inhibitors, including GapmeRs and siRNAs without the risks of dose-dependent toxicity such as hepatotoxicity, and the difficulty in targeting less accessible molecules predominantly localised to the nucleus [80]. Nevertheless, novel techniques to enable efficient delivery and tissue-specific enrichment of lncRNA-targeting molecules have been proposed, such as the use of viral vectors, exosomes, nanoparticles, cell-penetrating peptides, antibodies, and engineered extracellular vesicles [79,80,81,82]. Making use of on-target sequence specificity to enhance the efficient ncRNA delivery and reduce the unwanted off-target side effects has also been proposed [82], with the reported tools paving the way for disruption of lncRNAs with minimal off-target effects [81]. Moreover, poor evolutionary conservation of lncRNA represents an additional difficulty in their therapeutic development, requiring the critical evaluation of animal models used to ensure efficacy, and to avoid undesirable adverse effects [80]. 

## 7. Future Perspective and Directions 

Thus far, the advancements in research methods have significantly contributed to our understanding of lncRNA actions within the vasculature. Increasingly effective and sophisticated mechanistic studies are providing insight into the molecular mechanisms by which lncRNAs epigenetically regulate endothelial and VSMC gene expression in vascular homeostasis, ultimately enabling a more profound understanding of the pathological mechanisms involved in vascular disease and elucidating the potential therapeutic targets. The modulation of gene expression via targeting the epigenetic pathways implicated in angiogenesis, endothelial dysfunction, and vascular disease is a novel therapeutic avenue to explore in the future—however, greater characterisation of lncRNA interactions with the vascular gene promoters and epigenetic regulators such as PRC2 are required. Further advancements in the field largely depend on a better understanding of the structural organisation of lncRNAs and structure-to-function relations. Due to the poor conservation across species, the currently available tools for functional prediction based on a primary sequence similarity or structural predictions are not of great use [11]. However, the use of CRISPR (clustered regularly interspaced short palindromic repeats)-mediated gene editing to introduce targeted modifications into lncRNAs, together with novel technologies such as genome-wide chromatin interrogation, locked nucleic acids, bridged nucleic acids, and novel approaches to RNA purification in vitro and in vivo, will provide a more comprehensive understanding of the lncRNA structural domains, their folding patterns, and undiscovered mechanisms of action. 

## Figures and Tables

**Figure 1 ncrna-07-00062-f001:**
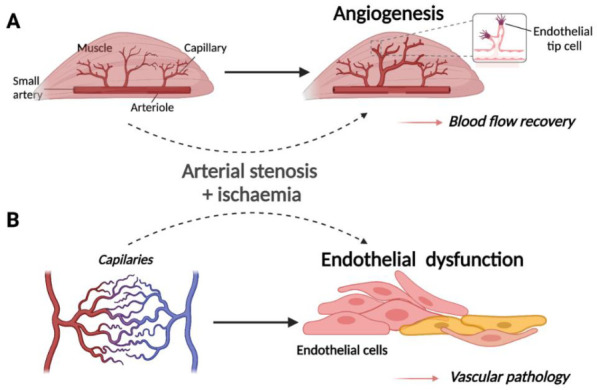
Schematic representation of (**A**) the normal physiological response to ischaemia in muscle tissue and (**B**) the pathogenesis of ischaemic vascular disease in endothelial cells (modified from [7]). Created with BioRender.com. (**A**) Upon restricted blood supply to the musculature, capillary beds within the ischaemic tissue undergo physiological angiogenesis whereby new vessels branch from the arterioles and infiltrate muscle fibres to increase the delivery of oxygen to metabolically active cells. (**B**) Upstream arterial stenosis or occlusion due to atherosclerosis may result in ischaemia of capillary ECs, promoting endothelial dysfunction and contributing to pathogenesis of occlusive arterial disease.

**Figure 2 ncrna-07-00062-f002:**
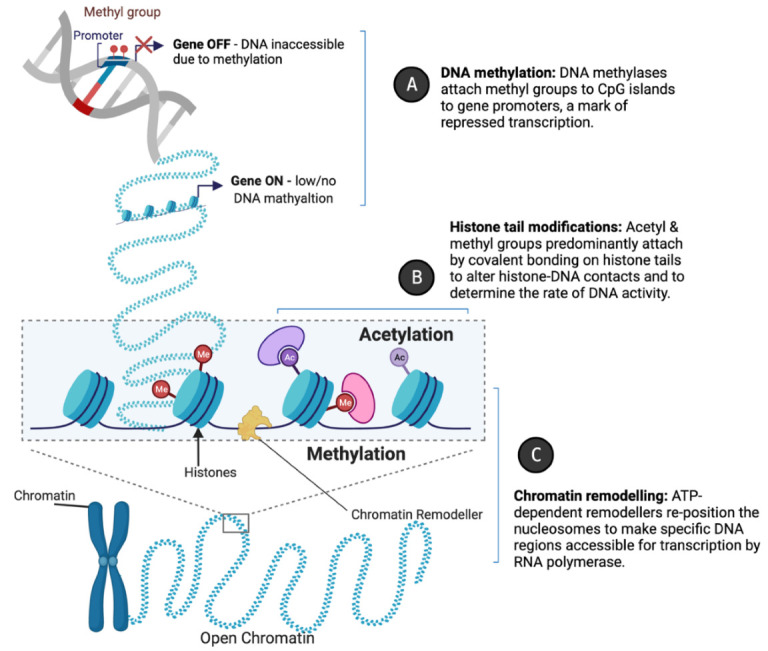
Epigenetic mechanisms of gene regulation. Created with BioRender.com. Epigenetic regulators such as (**A**) DNA methylases, (**B**) histone-modifying enzymes, and (**C**) chromatin remodellers modulate gene expression by mediating accessibility to DNA sequences for replicative enzymes or transcription factors. This is achieved through altering the compaction of chromatin by the enzymatic modification of DNA or histones, or by nucleosome repositioning.

**Figure 3 ncrna-07-00062-f003:**
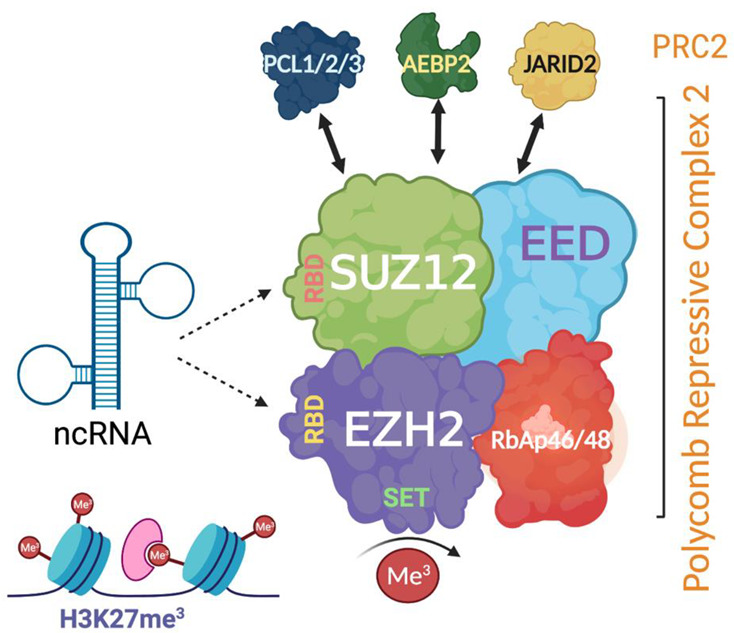
Schematic representation of the structure of polycomb repressive complex 2 (PRC2) and its interaction with non-coding RNA (ncRNA) via RNA binding sites. Created with BioRender.com. PRC2 has three major proteins including EZH2 (enhancer of zeste homolog 2), EED (embryonic ectoderm development), and SUZ12 (suppressor of zeste 12 homolog). EZH2 and SUZ12 have RNA-binding capacity. Accessory proteins such as JARID2 (jumonji and AT-rich interaction domain containing 2) associate and often co-purify with PRC2.

**Table 1 ncrna-07-00062-t001:** Epigenetic mechanisms of nuclear vascular lncRNAs and approaches used to explore their function.

lncRNA	Function	Epigenetic Mechanism	Cell Type	Ref
	Nucleus			
	**Chromatin isolation by RNA purification (ChIRP)**			
MEG3	Modulates activity of TGFb pathway genes by RNA–DNA triplex formation	Recruits PRC2 complex to repress TGFb by H3K27me3 methylation	BT-549	[26]
	**Chromatin immunoprecipitation (ChIP)**			
MALAT1	Mediates hyperglycaemia-induced glomerular EC injury and diabetic nephropathy	Recruits G9a methyltransferase to increase H3K9me1 levels on the *klotho* promoter to repress transcription	HRGECs	[27]
	**ChIP following lncRNA knockdown**			
NEAT1	Represses VSMC contractile genes to promote a proliferative phenotype and vascular repair in response to injury	Sequesters WDR5 modifier to decrease transcriptionally active chromatin marks (H3K4me3, H3K9ac) and increase inactive modifications, condensing chromatin structure and facilitating gene repression	HCASMCs	[22]
	**RNA immunoprecipitation (RIP)**			
LEENE	Promotes eNOS expression and endothelial cell function	Recruits and enhances binding of RNA polymerase II, initiating pro-transcriptional chromatin remodelling at the eNOS promoter	HUVECs	[21]
	**Individual nucleotide resolution UV cross-linking and immunoprecipitation (iCLIP)**			
MALAT1	Contributes to VSMC dysfunction in thoracic aortic aneurysm (TAA)	Binds BRG1 and histone deacetylase HDAC9 to pathologically target the complex to the nucleus where HDAC9 recruits EZH2 to transcriptionally silence VSMC-specific genes encoding contractile proteins	VSMCs	[28,29]
GATA6-AS	Regulates hypoxia-induced endothelial to mesenchymal transition (EndMT), angiogenesis and EC migration	Binds and negatively regulates H3K4me3 demethylase activity of LOXL2 chromatin modifier to increase activity of target genes	HUVECs	[30]
ANRIL	Contributes to development of atherosclerotic plaque through inhibiting macrophage reverse cholesterol transport (mRCT)	Recruits EZH2 and acts as a scaffold by forming RNA–DNA triplexes with the CDKN2B promoter to enrich H3K27me3 and induce repressive chromatin remodelling under regulation by CTCF protein to silence CDKN2B transcription	THP-1 macrophage-derived foam cells	[17,31]

BT-549—Human breast epithelial carcinoma cell line, HRGECs—Human renal glomerular endothelial cells, HCASMCs—Human coronary artery smooth muscle cells, HUVECs—Human umbilical vein endothelial cells, VSMCs—Vascular smooth muscle cells, BRG1—Brahma-related gene-1.

**Table 2 ncrna-07-00062-t002:** Total number of Gene Expression Omnibus (GEO) series reporting the use of named techniques in vascular field between 2018–present.

Search Terms		Reported
**Technique Used**	**Cell Type**	**Number of GEO Series**
RIP	Vascular cells	1 (GSE142386)
	Endothelial Cells	2
ATAC-seq	Endothelial Cells	13
	HAEC	4
CLASH	Endothelial Cells	1 (GSE101978)
iCLIP	Endothelial Cells	2
	HUVEC	1 (GSE99686)
ChIP-seq	HUVEC	23
	HAEC	2
	VSMC	2
	HPAEC	3
RNA-seq	Endothelial Cells	99
	HUVEC	44
	VSMC	3

RIP—RNA immunoprecipitation, ATAC-seq—Assay for transposase-accessible chromatin using sequencing, CLASH—Cross-linking ligation and sequencing of hybrids, iCLIP—Individual-nucleotide-resolution UV cross-linking and immunoprecipitation, ChIP-seq—Chromatin immunoprecipitation with sequencing, RNA-seq—RNA sequencing, HAEC—Human aortic endothelial cells, HPAEC—Primary human pulmonary artery endothelial cells.

## Data Availability

Not applicable.

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
