# Peer review of "Long Non-Coding RNA Regulation of Epigenetics in Vascular Cells"

_ncrna, 2021, doi:10.3390/ncrna7040062_

Round 1

Reviewer 1 Report

Garrattt et al. summarized our latest understanding of long non-coding RNA-based regulation of vascular homeostasis in health and in disease.  There are only a few minor points to improve the manuscript.  

Comments:

  1. This review article summarized the long non-coding RNA-based regulation of vascular homeostasis in endothelial cells and in smooth muscle cells. However, in the Abstract, the authors described the topics of endothelial cells only, but not of smooth muscle cells. The abstract should be modified. 

  1. Tables 1 and 3 are not sophisticated and should be modified.

  1. There are some typos in the manuscript.

Author Response

We are grateful to the Editor and both Reviewers for suggesting the changes. We believe we have significantly improved our manuscript now to include all suggestions. We have restructured the manuscript in a more logical order and included additional section

“6. Application and RNA therapeutics as per reviewer’s suggestion and in discussion with all the authors. Original Section 4 and 5 have been swapped and updated to “4. Polycomb Repressive Complex 2 and lncRNAs” and “5. Multi-omics approaches to detect long non-coding RNAs in vascular cells”.

We have expanded some of the sections as per reviewer’s 2 suggestion and discussed:

  • lncRNA secondary and tertiary structure
  • association between lncRNAs with extracellular vesicles and
  • diagnostic and prognostic value of circulating lncRNAs,

Further changes within the text have all been tracked. We have also included additional references.

There are now 2 Tables in total and 2 Figures. Figure 1 has been updated and re-drawn in BioRender. Tables 1 and 3 have been merged to represent the information they contained in a clearer and sophisticated way.

We have checked the manuscript for grammatical errors and edited as per native speaker’s suggestions. 

Response to Reviewer 1

  1. We are grateful to Reviewer1 for suggestions made to improve our manuscript. We have modified the part in the abstract to describe the topic of vascular cells including endothelial and smoot muscle cells.
  2. Tables 1 and 3 have been merged and modified to represent sophisticated info more clearly
  3. The manuscript has been checked for typos and grammatical errors and has been corrected as per native speaker’s suggestions. 

Reviewer 2 Report

1) The autors describe the importance of analyzing lncRNA secondary and tertiary structure. Since these structures relate to lncRNA function, this aspect should be better described and discussed.

2) The diagnostic and prognostic value of circulating lncRNAs is an open field of investigation. The association of lncRNAs with extracellular vesicles should be also discussed (For refs see also Kenneweg F. 2019).

3) In the light of the growing interest in the research on RNA therapeutics, the authors should discuss the development of potential future applications of lncRNA therapeutics (For refs see also Born LJ 2020, Battistelli C 2021, Winkle M 2021 and Lucas T 2018).

4) Some typing/grammar errors should be amended.

Author Response

We are grateful to the Editor and both Reviewers for suggesting the changes. We believe we have significantly improved our manuscript now to include all suggestions. We have restructured the manuscript in a more logical order and included additional section

“6. Application and RNA therapeutics as per reviewer’s suggestion and in discussion with all the authors. Original Section 4 and 5 have been swapped and updated to “4. Polycomb Repressive Complex 2 and lncRNAs” and “5. Multi-omics approaches to detect long non-coding RNAs in vascular cells”. 

We have expanded some of the sections as per reviewer’s 2 suggestion and discussed:

  • lncRNA secondary and tertiary structure
  • association between lncRNAs with extracellular vesicles and
  • diagnostic and prognostic value of circulating lncRNAs,

Further changes within the text have all been tracked. We have also included additional references.

There are now 2 Tables in total and 2 Figures. Figure 1 has been updated and re-drawn in BioRender. Tables 1 and 3 have been merged to represent the information they contained in a clearer and sophisticated way.

We have checked the manuscript for grammatical errors and edited as per native speaker’s suggestions. 

Response to Reviewer 2

Response to Reviewer 2 

  1. We have discussed and better described the importance of secondary and tertiary structure of lncRNAs
  2. The association of lncRNAs with extracellular vesicles has also been discussed and Kenneweg F. 2019 has been cited
  3. The diagnostic and prognostic value of circulating lncRNAs has been discussed
  4. The section on RNA therapeutics and future applications has been included.
  5. The manuscript has been checked for typos and grammar and edited as per native speaker’s suggestions. 

Round 2

Reviewer 2 Report

In my opinion the paper is suitable for publication